**Perspective**

# A soil food web approach to integrate soil fauna into multitrophic biogeochemistry
Justine D. M. Lejoly [1] ✉, Kyle Mason-Jones [1,2,3] & G. F. Ciska Veen [1]

Soil fauna is essential in mediating the formation and turnover of soil organic matter, the largest terrestrial pool of organic carbon, yet remains absent from nearly all existing biogeochemical conceptual frameworks. Soil fauna impacts element cycling via trophic (microbivory, herbivory, detritivory, predation) and non-trophic (bioturbation, dispersal, waste products) pathways. These effects should be integrated into multitrophic biogeochemistry by considering the environmental context (abiotic constraints, quality and quantity of carbon inputs), by including soil food web structure, and by accounting for the role of soil fauna as agents of connectivity. We highlight the need for better quantification of soil fauna and consideration of carcasses and feces as substantial carbon flows. Based on a soil food web approach, these concepts will improve the quantification of the impact of soil fauna on carbon cycling and help to predict how soil fauna will affect carbon cycling under future global change scenarios.

Soils store more carbon (C) than the atmosphere and vegetation combined[1] and can substantially contribute to climate change mitigation[2]. Soil microbes are the principal agents of soil organic matter (SOM) formation, involving the transfer and accumulation of organic C in the soil, and SOM persistence, its resistance to mineralization. These microbial effects occur through their decomposition of plant litter, assimilation of organic substances, and production of diverse microbial products[3,4]. Although soil fauna can be of equal importance in explaining variability in C and nitrogen (N) cycling processes[5] and their inclusion can improve SOM predictions[6], they have received much less attention than microbes[7].

Soil fauna is taxonomically and functionally diverse[8,9], and in most biomes surpasses aboveground faunal biomass[9]. Feeding strategies in soil food webs include microbivory, herbivory, detritivory, and predation of other fauna. Soil fauna, especially detritivores, substantially contributes to litter decomposition[10] while microbivores directly affect microbial biomass and activity[11], and root herbivores can alter the quality and quantity of C entering the soil[12,13]. Although typically considered small compared to microbial biomass[9], soil fauna has a high C turnover rate and may have disproportionate impacts on SOM formation and persistence by altering decomposition and nutrient cycling rates, as well as the fate of plant-derived C. These faunal impacts on biogeochemical cycles are already largely recognized in aboveground and aquatic systems as zoogeochemistry[14].

The consideration of soil fauna in biogeochemical cycles has been conceptualized[15–17], with three main mechanisms relevant for C cycling: plant residue consumption and transformation (i.e., detritivory),

bioturbation, and microbivory[15]. A recent conceptual framework identified fauna traits relevant to SOM pools, focussing on how detritivores and microbivores drive plant litter transformation and subsequent SOM formation[18]. However, soil fauna also has major impacts on C cycling via other pathways, such as herbivory and higher predation, as well as non-consumptive interactions between and within trophic guilds, which are largely not integrated into the most recent literature regarding soil fauna effects on SOM. Soil fauna is still absent from most soil carbon models and frameworks[19–21], or, at best, considered as a constant bioturbation coefficient[3,22].

Integrating soil fauna into our understanding of soil biogeochemistry is needed to develop more robust and accurate predictions of soil C cycling and storage, considering that global changes are altering soil food web structure with unknown consequences for biogeochemistry[23–25]. Here, we bring together current knowledge on the role of invertebrate soil fauna in regulating microbial processes and element cycling through trophic (related to consumption of a resource) and non-trophic (not-consumptive) pathways and identify overlooked mechanisms and pools. We propose to integrate soil fauna into our understanding of soil C cycling through a soil food web perspective while considering the environmental context (abiotic constraints, quality and quantity of C inputs) that they operate within. We highlight that soil fauna, including smaller organisms, act as agents of connectivity that can move C inputs, microorganisms, and SOM between soil compartments. We argue that standing biomass is insufficient to truly integrate the role of soil fauna in C cycling and that we need to quantify their

¹Department of Terrestrial Ecology, Netherlands Institute of Ecology, Wageningen, The Netherlands. ²Soil Microbial Interactions, Department of Geoscience, University of Tübingen, Tübingen, Germany. ³Cluster of Excellence (EXC 3121): TERRA – Terrestrial Geo-Biosphere Interactions in a Changing World, University of Tübingen, Tübingen, Germany. ✉e-mail: j.lejoly@nioo.knaw.nl

contribution via carcasses and excreta. In this perspective, we take conceptual and practical steps towards a meaningful inclusion of soil fauna into multitrophic biogeochemistry, anticipating that it can be used to predict their impact on C cycling under environmental changes.

## Soil carbon cycling and soil organic matter persistence

Soil C cycling involves a range of biological, chemical, and physical processes. Plants fuel this cycle with their C-rich litter (both above and belowground) and root exudates. The decomposition, transformation, and/ or transport, of these plant inputs contribute to SOM formation while sustaining soil life[26,27]. Microbes can directly utilize labile plant C such as root exudates and litter leachates to produce biomass, extracellular enzymes, storage compounds, as well as extracellular polymeric substances[28]. When microbes die, their necromass can accumulate in soils and substantially contribute to soil C storage. In temperate grassland and arable soils, microbial necromass can account for more than 50% of soil organic C[29,30]. Meanwhile, litter can be fragmented by detritivorous fauna, which accelerates its decomposition[10]. These biological transformations of plant inputs have a dual effect on C cycling: they contribute to persistent SOM formation and to $CO_2$ production through respiration.

The persistence of SOM can be functionally understood by its partitioning into three pools of contrasting physicochemical composition and formation pathways. Particulate organic matter (POM), mostly consisting of partly decomposed plant fragments, has a mean residence time of years to decades and is not protected from decomposition unless it is incorporated into aggregates (occluded POM)[31]. Mineral-associated organic matter (MAOM) contains smaller molecules of both plant and microbial origin associated with minerals and is considered more persistent than POM, with mean residence time up to centuries[31,32]. Root exudates and litter leachates make important contributions to MAOM formation, either by direct sorption to silt and clay particles or through microbial transformations and necromass[19,33,34]. Dissolved organic matter, a smaller pool than MAOM and POM, contains unprotected small organic molecules including exudates and leachates and represents a dynamic precursor of MAOM[34,35].

Broadly speaking, the relative contribution of POM and MAOM to soil C storage is dependent on climate and land use[34,36]. When biological activity is limited by low temperatures or moisture, SOM mostly accumulates as POM[34]. For example, in boreal forests where temperatures are lower, the contribution of MAOM to total SOM is lower (45%) than in temperate forests (64%). Meanwhile, tropical forests accumulate very little POM (<10%), due to very high decomposition rates[34]. Different land use types are associated with contrasting plant input quality (e.g., C:N ratio, lignin content) and quantity. Because of generally high plant input quality, grassland soils have a high proportion of MAOM (70%) and as do arable soils, although SOM concentrations are lower overall in the latter. Within the same land use type, differences in plant input quality can also affect the distribution between POM and MAOM. Broadleaf forests typically have higher litter quality—notably lower C:N ratios—than coniferous forests and therefore have higher MAOM proportions[36].

In forests, these differences in litter quality, together with pedological conditions, also translate into humus forms[37]. Humus here refers to the organic layer on the soil surface where litter accumulates[37]. When litter is of high quality, soil fauna is abundant and incorporation of aboveground litter into the soil is fast. This leads to the development of mull humus forms, with a high proportion of soil C stabilized as MAOM[36]. On the other end of the spectrum, when conditions are suboptimal for litter decomposition and macrofauna, litter accumulates on the surface, corresponding to mor humus forms[37]. In between these two extremes, we find moder humus forms, typically dominated by ectomycorrhizal fungi and detritivores and where C is stored as POM in the mineral soil[36,37].

Current research mostly focusses on MAOM and on microbes, whose role in SOM formation and turnover is widely recognized[3,4,29,30,38]. However, soil fauna is likely to impact the formation and persistence of all three SOM pools (MAOM, POM, dissolved organic matter) and might also stimulate C transfer from POM to MAOM, while affecting other biogeochemical

processes. Recent reviews conceptualized the effects of different fauna taxonomic groups on SOM formation primarily through the litter pathway[15] and using functional traits relevant to organic matter transformations[18]. While this approach can help formulate the impacts of individual fauna groups on SOM pools using existing literature, it does not account for interactions between these groups in complex soil food webs nor does it consider C cycling beyond SOM formation. Beyond quantifying SOM pools, a comprehensive vision of soil C cycling should include heterotrophic respiration, litter decomposition, and microbial transformations, as well as modification of these processes through trophic interactions and non-trophic pathways. By starting from a soil food web perspective and separating soil fauna effects into different trophic and non-trophic pathways, we argue that we can gain better insights into their impacts on multiple C cycling processes and truly integrate them into multitrophic biogeochemistry.

## Trophic pathways in soil food webs

**Microbivory.** Soil microbes (e.g., bacteria and fungi) are key constituents of soil food webs and are prey to microbivores, including bacterivores and fungivores, which can control their community composition, abundance, and activity[39–41]. Bacterivorous nematodes and protists typically ingest the entire bacterium[42], while fungivorous nematodes, collembola, and oribatid mites, feed on living fungal hyphae[43,44]. Bacterivores decrease soil microbial biomass by 16% on average, while stimulating microbial respiration and turnover, which can increase C and N mineralization[11,41,45,46]. A recent study manipulating protist size revealed that only larger bodied species significantly decrease litter decomposition while increasing C mineralization[41]. Meanwhile, fungivores can accelerate organic matter decomposition and decrease fungal-induced soil aggregation, directly influencing SOM persistence and C cycling[39,43,44,47]. When feeding on plant symbionts or pathogens, microbivores can also affect plant growth, thereby altering plant C inputs[11,48,49].

The impacts of bacterivores and fungivores on microbial processes are often studied separately although bacteria and fungi extensively interact and compete for resources[50]. We currently lack a comprehensive understanding of the combined impact of bacterivores and fungivores on C cycling, which we expect depends on the relative dominance of fungi and bacteria.

**Detritivory.** Detritivores, including isopods, snails, and millipedes, feed on plant litter and control resource availability for microbial decomposition through fragmentation[51]. Detritivory can accelerate C cycling by 38% compared to microbial decomposition alone, by increasing C availability to microbes (reducing resource C:N ratios and increasing dissolved C and N[52]). Soil fauna has a significant effect on leaf litter decomposition, ingesting on average 49% of annual litterfall and up to 100% in temperate grasslands[10,53]. Their contribution to litter decomposition increases with moisture content[54,55], peaking in wetter and warmer biomes[10]. They also increase root litter[56,57] and wood[58] decomposition. By fragmenting and transforming litter, detritivores have a direct impact on POM formation[31,59] and can contribute to MAOM formation via litter leachates[60,61].

The impact of different detritivores on C cycling is likely to differ based on feeding traits. For example, we would expect the transfer from POM to MAOM to be slower for detritivores exclusively feeding on litter or POM, in contrast with those simultaneously feeding on soil and directly mixing organic matter with the mineral matrix. Notably, earthworms can accelerate the conversion of labile plant C into microbial necromass[62] and the transfer from POM to MAOM[63]. No similar studies were done on other detritivores, whose assimilation efficiency (how much of ingested food is not excreted) can greatly vary, ranging from 10 to 83%[64–67], and is likely directly proportional to the strength of their impact on C cycling, higher assimilation efficiency corresponding to more C transformed.

Detritivores are relatively well studied in the context of litter decomposition and recent literature is starting to link them to other C cycling processes, notably studying chemical changes associated with litter-to-feces

transformation[60]. However, consequences for SOM persistence remain hypothetical and few studies have looked at their relation to POM and MAOM[15].

**Predation.** Predators exert top-down controls on soil food webs and associated processes through multiple mechanisms. By regulating their populations through consumption, predators can have positive effects on prey diversity and activity[68,69]. By altering the behavior and energetic requirements of their prey, they can also affect lower trophic levels through trophic cascades[70–72]. Soil predators include carnivorous nematodes, spiders, predatory mites, centipedes, and pseudoscorpions. Pseudoscorpions and predatory mites can increase microbial and fungal biomass, by decreasing fungivore abundance[73,74]. In some cases, these changes in microbial community composition in turn altered C and N cycling, although the strength and direction of these trophic cascades vary.

A few manipulative studies found that pseudoscorpions and predatory mites did not affect C mineralization[73–75], except at very low fungal diversity where it increased[68]. Similarly, a study found that spiders did not affect litter decomposition, regardless of predator density[70]. Although other C cycling parameters are rarely investigated, a study found that pseudoscorpions increased microbial efficiency and decreased POM C:N ratios[73]. Predatory mites can also affect N cycling, increasing $N_2O$ emissions through increased nitrification[72].

The effects of predators depend on environmental conditions and resource availability. A study found that predatory mites increased fungal biomass by controlling collembola abundance only in N-poor conditions, suggesting prevalent top-down control under nutrient-limiting conditions[74]. Meanwhile, another study found that predatory mites had a stronger top-down control over collembola with higher plant diversity[76]. Warming might increase top-down controls of predators by increasing their metabolic requirements, although not all predators are equally affected. Predatory mites were found to be more sensitive than centipedes to a warming-drought combination[77]. While microbivores and decomposers are integrated in recent SOM conceptual frameworks[15,18], higher predators have never been explicitly considered, to the best of our knowledge. Literature on higher soil predators and their effects on element cycling is scarce and there is an urgent need for additional experimental research on multiple predator taxa.

**Root herbivory and additional trophic pathways.** Root herbivory can increase root C allocation and alter the composition of root exudates, with consequences for microbial activity, as well as plant growth and community composition[12,13,78–82]. At low density, root-feeding nematodes can notably increase microbial activity in the rhizosphere[12]. Herbivores consume on average 4% of fine root biomass[53], which represents less than 10% to basal resources of soil food webs[26,27]. Some root herbivores directly feed on xylem and phloem (known as root suckers) without ingesting root fragments[79] and are likely to alter root development and exudation patterns, although their contribution to C fluxes is unknown. Plant pathogens such as oomycetes are also likely affecting element cycling but, to the best of our knowledge, evidence is lacking.

It is estimated that up to 20% of described soil faunal species feed on non-vascular plants, including lichen, bryophytes, and algae[83], which are important primary producers at high altitudes and latitudes, and in biocrusts. Studies on interactions between non-vascular plants and soil fauna are scarce, although lichenivorous microarthropods were found to increase lichen decomposition[84]. More research on how fauna feeding on non-vascular plants and on plant roots influences element cycling is needed.

## Non-trophic pathways affecting carbon cycling

**Bioturbation.** Bioturbation is the only process performed by soil fauna currently represented in some biogeochemical models[3,22]. However, bioturbation is more than the vertical transfer of POM to lower soil horizons[22]. Soil macrofauna can modify soil structure, via biopore creation, increased pore connectivity, and aggregation[85].

Bioturbation by soil macrofauna can have long-lasting effects on the soil physicochemical environment, altering soil abiotic properties[86,87], nutrient availability[88], pH[89,90], aggregation[91,92] and water infiltration[93]. A global analysis found that major soil macroinvertebrates (earthworms, ants, and termites) all increase soil macronutrient content, soil respiration, and biomass of plants and microorganisms[21]. Termites increase clay content while earthworms and ants increase and decrease moisture content, respectively[21]. These physicochemical changes directly affect C cycling[92,94–96] and modify the soil as a habitat for organisms[97] and plants[98].

The role of these soil invertebrates in shaping landscapes is recognized but likely underestimated compared to larger animals[99]. Termites and ants create heterogeneity by building long lasting mounds[88,100] and/or temporary sheetings to ensure access to food resources[101], horizontally redistributing elements. Although bioturbation is the most studied process by which soil fauna affects C cycling, it is unclear how it interacts with other trophic and non-trophic pathways.

**Excreta.** All animals produce excreta, in the form of feces and/or urine, but also mucus for earthworms, enchytraeids, and snails[102]. These excreta are hotspots for microbial activity and substantial pools of available nutrients which can locally increase plant nutrient uptake[103]. Feces are rich in C from partly digested litter fragments, contributing to POM, and also contain substantial amounts of dissolved organic matter. Detritivore feces are also rich in dissolved N[52] and can increase N leaching and total N soil content[104,105]. High nutrient and C availability can alter litter and SOM decomposition, notably through priming effects[105,106]. Meanwhile, earthworm mucus was found to promote MAOM formation[107]. Most of these experimental studies were done on detritivorous macrofauna (but see ref. [105]), while there is little information about the contribution of soil fauna excreta to C cycling beyond detritivore feces and macrofauna[108,109].

**Carcasses.** Across ecosystems, carcasses are rich in nutrients and have low C:N ratios compared to plant material, making them a readily available source of energy. Following carcass density-size relationships established for terrestrial and aquatic systems, smaller soil organisms should have higher carcass densities[103]. Belowground, only earthworm carcasses have been experimentally studied and were shown to alter plant and microbial community composition, as well as increase plant biomass compared to living earthworms[110,111]. Additionally, soil arthropods have a chitin-rich exoskeleton that they regularly shed (once every 21 days for the collembola *Lepidocyrtus curvicollis*[112]). As a result, exoskeleton carcasses may exceed standing biomass[113]. While we expect carcasses and exoskeletons to represent substantial pools of C and nutrients, their contribution to element cycling is, to the best of our knowledge, unknown.

**Additional non-trophic pathways.** Different species or organism groups can also interact through non-consumptive mechanisms, including facilitation, competition and habitat modifications. By feeding on plant litter, isopods were found to alter nematode community composition, although field and laboratory settings yielded contrasting outcomes[114]. Effects on soil C storage are also inconclusive. A recent study found antagonistic effects of multiple earthworm species on POM persistence, likely due to resource competition[115], while another study found synergistic effects on C stocks, explained by higher cast production[96]. Through non-trophic interactions, multiple soil fauna groups can also influence greenhouse gas emissions: higher species diversity was associated with increased $CO_2$ but decreased $N_2O$ emissions[116]. These findings come from a limited number of experimental studies and highlight the need for additional studies of the effects of non-trophic interactions between and within feeding groups on C cycling.

**The environmental context of multitrophic biogeochemistry**
**Fauna should be considered in the context of their abiotic constraints.** The influence of soil fauna on C cycling is highly dependent on abiotic constraints, including aboveground (micro)climatic conditions and belowground soil physicochemical properties. These abiotic constraints determine to a large extent which species can thrive in a given environment but are rarely explicitly recognized in conceptual frameworks linking soil fauna to biogeochemistry[15,18].

Climate and soil properties influence the composition of soil fauna communities, with taxon-specific responses. Globally, earthworms and collembola are primarily driven by climatic variables[117,118], while nematodes are more strongly associated with soil properties[119]. Earthworms are locally driven by nutrient availability and pH, the latter being an even stronger driver in acidic soils[90]. The net effect of soil fauna on C storage can be influenced by soil age, with earthworms promoting MAOM formation in younger soils but increasing C mineralization in older soils[120].

Soil fauna is also influenced by physical properties, including pore size and connectivity, as well as aggregation, soil structure notably affecting prey-predator interactions[121,122]. As water availability is a determinant for the

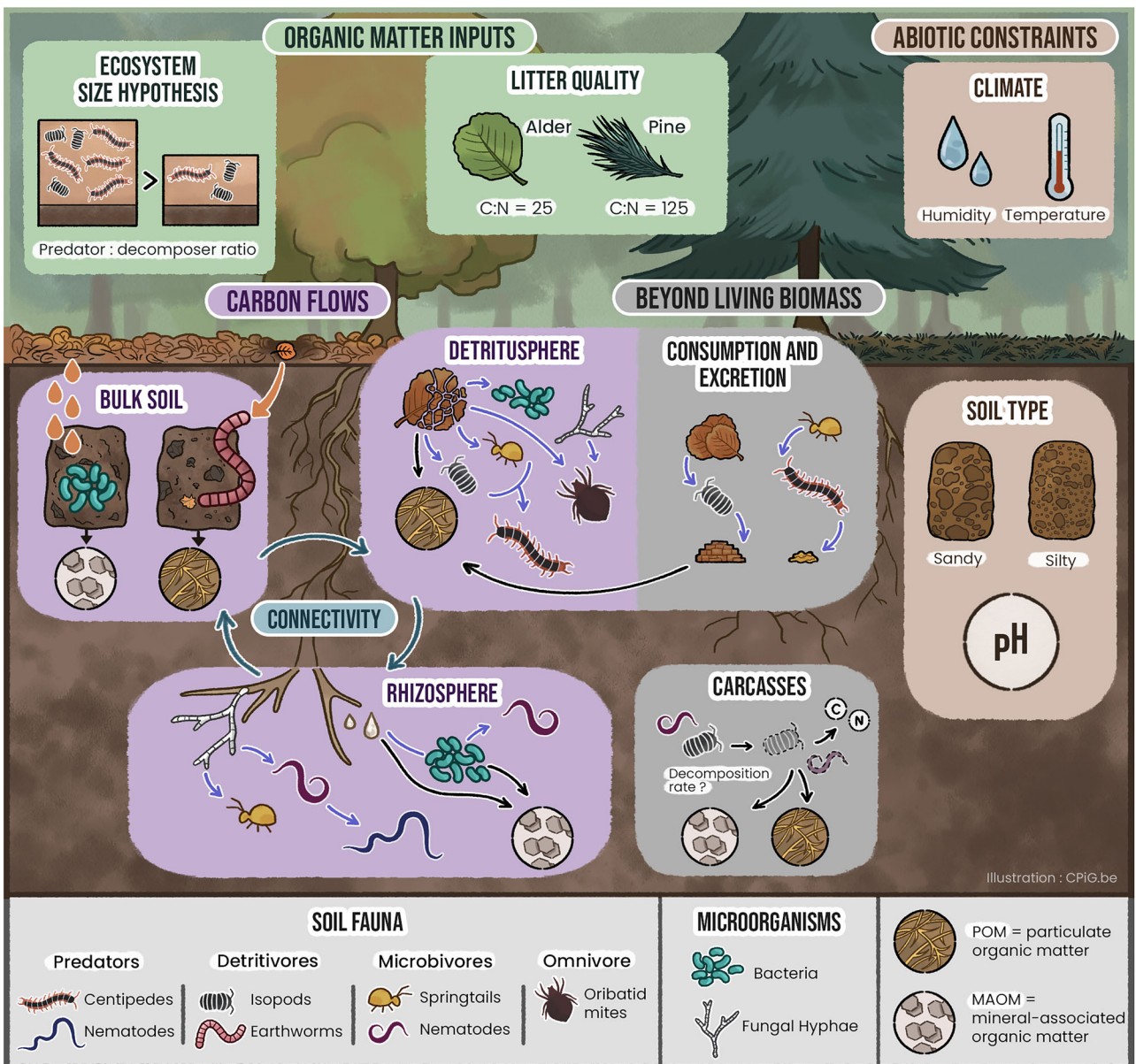

**Fig. 1 | Five key concepts to integrate soil fauna into multitrophic biogeochemistry.** This figure illustrates pathways by which soil fauna can influence soil carbon (C) cycling and contribute to particulate organic matter (POM) and mineral-associated organic matter (MAOM) pools: (1) The abiotic constraints for soil fauna and SOM formation pathways are represented in the 'Climate' and 'Soil type' panes; (2) the importance to consider quality and quantity of C inputs is showed in the 'Litter quality' and 'Ecosystem size hypothesis' panes, as well as in the different soil compartments associated with contrasting plant inputs (the rhizosphere primarily associated with root exudates, the detritusphere with litter fragments, and the bulk soil with sporadic inputs from litter leachates (orange drops) and bioturbation; (3) the role of soil fauna feeding preferences and how C flows through the soil food web and into SOM pools is illustrated in different soil compartments, of which (4) soil fauna increases the connectivity; and (5) soil fauna pools beyond living biomass are presented in the 'Carcasses' and 'Consumption and excretion' panes. The latter also links to important feeding traits: a detritivore such as an isopod ingests a great amount of litter, most of which is excreted as feces, because of lower food quality and assimilation rates, while a predator such as a centipede comparatively ingests much smaller amounts of prey, as a result of higher food quality and assimilation rates. Purple arrows indicate C flows through living organisms and black arrows after-life transfers of C and nutrients into SOM pools. The figure was entirely designed by CPiG (Carolina Levicek), including all individual elements.

survival and movement of aquatic organisms such as nematodes and protists, extremely dry and wet conditions can both negatively affect biological activity. Soil fauna can modify soil physicochemical properties to some extend (see section "Bioturbation"). Because these abiotic conditions also influence C cycling (see section "Trophic pathways in soil food webs"), aboveground and belowground environmental conditions need to be included in multitrophic biogeochemistry (Fig. 1).

**Carbon on the menu: the role of soil fauna is dependent on C inputs.**
The impact of soil fauna on C cycling is directly connected to the C used as a resource. Current conceptual frameworks view plant C inputs as a continuous pool ranging from labile (e.g., root exudates) to recalcitrant (e.g., coniferous needles), with labile C resulting in greater stable SOM[3,50], but do not yet explicitly link soil food webs with C cycling. Because of variations in stoichiometry and chemical complexity, the consequence of feeding interactions for C cycling must be examined in the context of available C resources, providing a strong basis for multitrophic biogeochemistry (Fig. 2).

Plant inputs vary spatially in the soil, from labile root exudates in the rhizosphere to plant litter in the detritusphere and sporadic litter leachate in the bulk soil, especially in preferential flow paths[123] (Fig. 1, 2). Microbivores are abundant in the rhizosphere and can enhance microbial activity and C use efficiency[8], regulating the ability of microbes to transform labile C inputs into MAOM[8,124,125]. Root exudates mostly enter the soil food web through microbes, but can be quickly (within days) assimilated by microarthropods and even earthworms[126]. In the detritusphere, detritivores play a predominant role in litter decomposition[37,127] and directly contribute to POM formation. Litter inputs in the rhizosphere can also be substantial, through root turnover and herbivory[79]. Both litter and root exudates promote soil fauna[128] while lower faunal and microbial biomass is found in the bulk soil[123], where impacts of soil fauna on SOM formation and persistence are likely smaller.

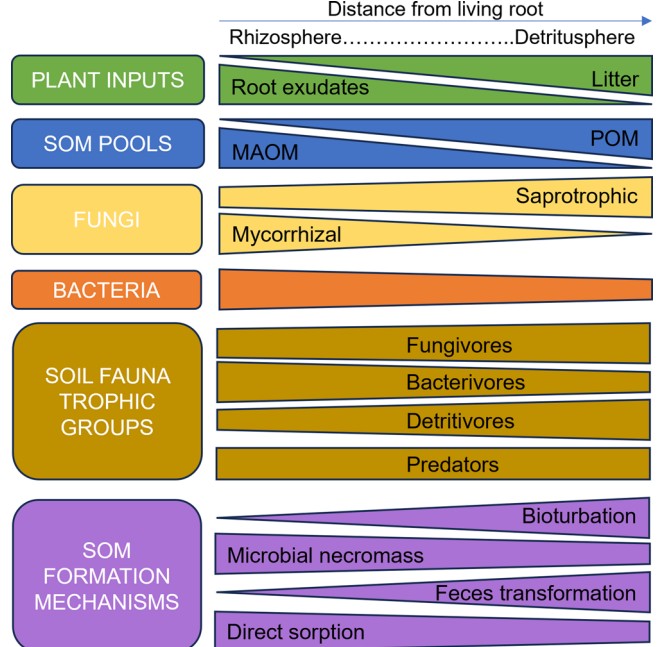

**Fig. 2 | Conceptualization of the role of different soil fauna trophic groups under contrasting plant inputs.** The quality and availability of plant inputs can be seen as a continuum from the rhizosphere to the detritusphere associated with contrasting soil faunal and microbial communities. The three-way interaction between plant carbon inputs, soil microbes, and soil fauna determine the dominant mechanisms for soil organic matter (SOM) formation and the dominant SOM pools (particulate organic matter, POM, and mineral-associated organic matter, MAOM), given that abiotic conditions are kept constant.

Soil fauna can modify the fate of C inputs through chemical changes[60,129] and influence their relative contributions to POM versus MAOM formation pathways[120,130]. Litter fragmentation by detritivores releases dissolved organic molecules and increases microbial access, thus creating a pathway to MAOM formation through direct sorption and microbial transformations such as necromass formation[60,131]. However, these effects are also dependent on litter quality and land use[60,120,129,132].

In forest soils, the thickness of organic horizons (humus) can be a determinant of fauna community structure, as it is an important habitat for many species. Forests with thicker and less disturbed organic horizons are associated with higher proportions of predators, both for soil arthropods[133] and carabid beetles[134]. These observations follow the ecosystem size hypothesis, which states that higher trophic levels, i.e., predators, are limited by habitat size rather than prey availability[135]. This principle could be used to predict soil food web structure and the relative importance of higher trophic levels.

## The core of multitrophic biogeochemistry

**Soil food web structure determines carbon flows.** Not all soil fauna is the same and all soil fauna deserves to be integrated into biogeochemistry. We propose to start the conceptualization of multitrophic biogeochemistry from a soil food web perspective, by considering all feeding guilds and trophic interactions. We argue that more than directly affecting total soil C pools, soil fauna can change the fate of C, its availability, and its cycling rates. Considering the dynamic nature of soil life and associated C transformations, it is essential to shift from a C pool to a C flow perspective.

Different trophic guilds have contrasting effects on C flows and SOM persistence (Fig. 1). Fungivores and bacterivores generally increase soil C mineralization, thereby potentially decreasing SOM persistence (Table 1). On the other hand, microbivores may increase microbial necromass formation by accelerating microbial turnover[11] and C uptake[136], potentially increasing SOM persistence. The net effect on soil C storage is therefore complex to predict. Meanwhile, detritivores increase litter fragmentation, increasing its availability for microbial decomposition and its ability to act as a nucleus for aggregate formation[38,137], but potentially also increasing microbial necromass decomposition through ingestion of partly decomposed plant litter and POM[29,130]. Compared to other faunal groups, there are more studies on earthworms[15], which speed up MAOM formation[63] and microbial necromass formation from labile C inputs[62]. The role of predators in SOM formation and persistence is more complex to predict, especially as they also act through indirect trophic pathways (e.g., trophic cascades). We expect predators of microbivores to generally enhance SOM persistence by mitigating the negative effects of microbivores, while predators of detritivores would slow down decomposition and stable SOM formation. However, omnivory can complicate predictions of soil faunal effects on C cycling (Box 1).

In addition to feeding groups, other factors play a role in determining the impact of soil fauna on SOM persistence[18,138–140]. Relevant feeding and growth traits were recently identified and comprehensively linked to SOM transformation processes[18]. However, determining the direction of their impact on C cycling can be challenging, especially as it can be density dependent[141]. At intermediate and low rates, predation generally has positive effects on prey density by reducing intraguild competition, but at higher rates, this effect can become negative. Lower feeding intensity of bacterivorous nematodes was notably associated with a greater increase of microbial biomass compared to higher feeding intensity[142]. This density-dependence was also confirmed for fungivores[143–145] and predatory mites[146]. Future research should investigate whether relationships between traits and C cycling or abundance and C cycling are linear or hump-shaped.

**Soil fauna as agents of connectivity.** Conceptually distinct soil compartments (detritusphere, rhizosphere, and bulk soil) in reality represent a continuum of environmental conditions (Fig. 1). The activity of soil fauna enhances connectivity between these compartments as mobile

**Table 1 | Soil fauna impacts on microbial processes, carbon (C) and nitrogen (N) cycling, and soil organic matter (SOM) persistence**

| Trophic group | Microbial processes | Element cycling | SOM persistence |
|---|---|---|---|
| FUNGIVORE | ↑ enzyme activity[178] | ↑ decomposition[47]<br>↑ soil N content[105] | ↓ aggregation[44]<br>↓ stabilization of fungal necromass[47] |
| BACTERIVORE | ↑ microbial turnover[46] | ↑ N mineralization[5,46]<br>↑ C mineralization[46] | ? |
| DETRITIVORE | ↓ microbial activity[179]<br>↑ / ↓ microbial biomass[179,180]<br>↑/= enzyme activity[181]<br>↓/= fungal dominance[181] | ↑ decomposition[137,182]<br>↑ C mineralization[67]<br>↑ N availability[179]<br>↑ soil N content[105]<br>↓ C:N ratios of C inputs (feces)[60] | ↑ microbial necromass formation[62]<br>↑ C protection in aggregates[85]<br>↑ MAOM formation[63] |
| PREDATOR OF MICROBIVORE | ↑ metabolic activity[73,75]<br>↑ microbial efficiency[73]<br>↑ enzyme activity[73]<br>↑ fungal dominance[73] | ↑ C mineralization[68]<br>↑ N mineralization[72] | ↓ C:N of POM[73] |
| PREDATOR OF DETRITIVORE | ? | ? | ? |
| ROOT HERBIVORE | ↑ microbial activity[79] | ? | ↑ SOM precursors[79] |

Unknown impacts are depicted as a question mark.
*MAOM* mineral-associated organic matter, *POM* particulate organic matter.

## Box 1 | How omnivory can affect predictions of soil fauna effects on C cycling

Omnivory is common in soil food webs, with most soil fauna species feeding on multiple resource types and harbouring diet plasticity[183–185]. Many fungivorous microarthropods, namely collembola and oribatid mites, can also be classified as detritivores, as they are also feed on decaying plant material[184]. The level of omnivory of these microarthropods can depend on resource availability and climatic conditions: lower-quality litter is associated with higher omnivory[186] and diet plasticity is greater in tropical than in temperate forests[165,187,188]. An omnivorous species can impact multiple C flows and its net effect on C cycling may be more difficult to predict compared to an obligate bacterivore. Diet plasticity also determines how well a species can adapt to disturbances and global changes by switching between resources based on their availability. A study found that the trophic position of microarthropods increased with altitude, likely resulting from the associated decrease in litter quality[189]. Additionally, as omnivore abundance is not purely dependent on a particular resource, pressure from generalist predators may be high even with low prey abundance.

links[147], with macrofauna having the largest impact through bioturbation (see section "Bioturbation") and biggest body size. However, smaller organisms (meso and microfauna) also act as agents of connectivity at different scales.

Soil fauna increases transfer of C inputs between compartments. Plant parasitic nematodes notably increase the amount of newly photosynthesized C that can be found in the bulk soil[79]. Detritivores transform and relocate plant inputs, thereby enhancing connectivity between detritusphere and other soil compartments[52] and increasing input of SOM precursors in the mineral soil[37]. Soil fauna can therefore alter the nutrient balance in soils by moving resources between soil compartments. An example with far-reaching consequences is the invasion of earthworms into northern forests, associated with a shift from mor or moder to mull humus forms, drastically modifying the existing equilibrium by increasing bioturbation (Box 2).

Soil fauna also facilitates the dispersal of micro-organisms[51]. Earthworms and microarthropods can transport bacteria and fungal spores through ectozoochory and endozoochory[148–151], while nematodes can transport viruses[152]. These fauna-driven movements of microbes and viruses might be key in shaping microbial community composition and co-occurrence networks[153,154].

### Better quantification of soil fauna pools is needed to implement multitrophic biogeochemistry

To fully understand the impact of soil fauna on C cycling, we need robust biomass data and the quantification of other important pools and fluxes. Soil fauna is often quantified using abundance, that is number of individuals per unit area, volume or soil mass. While this is an easy metric to obtain, it is only indirectly linked to biogeochemical processes. Biomass is a more relevant metric to quantify the impact of different soil organisms on these processes. Obtaining robust biomass estimates is challenging but should be a priority to accurately quantify the contribution of soil fauna to C cycling.

However, across diverse taxa, standing biomass is not proportional to activity and may not be the most appropriate proxy of fauna-driven C fluxes. The relevance of biomass to understand and quantify the role of biota in soil functions has already been questioned for microbes, for which growth-related measurements exist[155]. A stable isotope probing experiment revealed that microbial biomass C pool size is not proportional to its effect on C cycling, with a smaller fungal C pool but higher incorporation compared to bacteria[33]. There is also empirical evidence that microbial stoichiometric requirements predict the effect of C and N addition better than microbial biomass[156]. For soil fauna, alternatives to biomass measurements remain scarce, but soil faunal stoichiometry is worth exploring as a relevant metric for quantifying and predicting their effect on soil C cycling[157–159].

Beyond living biomass, soil organisms produce feces and carcasses whose contribution to SOM pools and C fluxes is unknown and not explicitly acknowledged. Microarthropod and enchytraeid feces are diagnostic for the identification of zoogenic humus forms in forests[160], yet their quantification is missing. Twenty years after a call for an estimation of feces production rates and research on their fate[161], these questions still remain mostly unanswered. Feces and carcasses, including exoskeletons, are likely to contribute substantially to SOM but represent a major knowledge gap for multitrophic biogeochemistry. Integration of these waste products in future frameworks is key to properly estimate the role of soil fauna in element cycling and shift from a pool to a flux perspective.

## Box 2 | Drastic changes in C cycling associated with earthworm invasion in northern ecosystems

Although differentiation between humus forms is generally believed to be driven by plant C quality and therefore vegetation (section II), a shift from moder or mor to mull can be observed after earthworm invasion[89], with far reaching consequences for element cycling (Box Fig. 1). By increasing the mixing between organic and mineral soil horizons, earthworms were found to promote fast growing bacteria and decrease C stocks and N limitations[89,190,191]. By decreasing the thickness of soil organic layers, they also alter understory vegetation and decrease seedling success[192] while reducing the abundance and diversity of both soil (meso and micro) fauna[193,194] and aboveground arthropods[195]. The decrease in soil C storage observed with earthworm invasion does not align with most laboratory experiments finding positive impacts of earthworms on SOM persistence[62,85,96,196]. This highlights the importance of confirming laboratory studies in observational field studies and proves that soil fauna alone can drive C cycling and microbial activity in the field without changes in plant C quality.

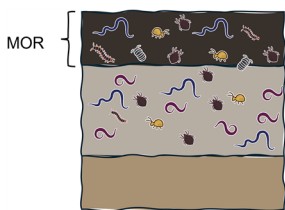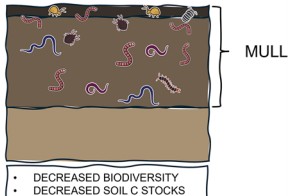

MOR    MULL

- DECREASED BIODIVERSITY
- DECREASED SOIL C STOCKS

**Box Fig 1. Impacts of earthworm invasion on soils.** The figure was created using soil organism sketches designed by CPiG (Carolina Levicek).

## Moving forward

Current frameworks for disentangling the drivers of soil C cycling and SOM persistence are missing the key element of soil fauna[15,18,21]. In biogeochemical approaches, the *bio* is generally limited to plants and microbes. Manipulative experiments investigating the biotic drivers of C cycling mostly exclude or do not account for soil fauna, which can lead to unrealistic findings. Meanwhile, experiments considering the role of soil fauna in C cycling are often limited to litter decomposition[10], insufficient to understand their impact on soil C storage[37,161]. We therefore encourage future litter decomposition experiments to measure additional parameters related to C cycling (heterotrophic respiration, microbial necromass, MAOM and POM pools), e.g., by using isotopic labeling or litter boxes instead of litter bags[162], and consider chemical transformations of litter by soil fauna. As most soil fauna simultaneously influences multiple biogeochemical pathways – sometimes in opposite directions –, the net effect for soil C storage and SOM persistence may not be easy to infer. We advocate for a better integration of multiple C cycling parameters in multitrophic soil biogeochemistry.

Because correlation-based approaches are insufficient to truly quantify the impact of soil fauna, we call for more experiments manipulating the composition of the soil food web and investigating the conditions under which soil fauna communities may or may not substantially affect C cycling[74,141]. Priority should be given to under-studied biomes, especially boreal regions, storing large amounts of C belowground and particularly susceptible to climate change[163,164] and (sub)tropical regions where biological activity is high and dominant fauna traits may differ from those of temperate regions[21,27,34,99,165]. Combining soil food web manipulation with stable isotope probing to trace C or N into different biomass and SOM pools would be groundbreaking for the development of multitrophic biogeochemistry and a better focus on C fluxes.

Global changes are impacting soil C cycling and soil biota. Warmer temperatures increase the direct contribution of soil fauna to energy fluxes[166] and to litter decomposition[167,168], but may decrease soil fauna abundance in the long-term[169]. Effects of climate change can also be dependent on land use intensity, as a study found that lower intensity was associated with greater contribution of microbivores to energy fluxes under warming and summer drought[170]. Meanwhile, elevated $CO_2$ can positively impact soil fauna, especially detritivores, and microbial activity, supposedly through increased primary productivity[171,172], but drier conditions can decrease soil fauna abundance and alter predator-prey interactions[173,174]. However, the impact may depend on the trophic group considered[172,175]. Larger and predatory soil organisms are disproportionately affected by agricultural intensification[24,25] and forest disturbances[176], while range expansion and human activities can alter soil food web composition[177]. All these global change factors can greatly reshape the composition and functioning of soil biota, with largely unknown consequences for element cycling. Research is needed to elucidate how the relationships between soil fauna and C cycling will evolve in a changing world.

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

## Acknowledgements
J.D.M.L. acknowledges funding by the European Research Executive Agency (REA) for the Marie Skłodowska-Curie Actions Postdoctoral Fellowship Project no. 101105509 (Soil Fauna MIND). G.F.V. acknowledges the Dutch Research Council (NWO) for funding of the Aspasia project 15.015.031. K.M.J. acknowledges the Dutch Research Council (NWO) for funding of the Veni project VI. Veni. 202.086 and the German Research Foundation (DFG) for the Emmy Noether project 521889691. Views and opinions expressed are however those of the author(s) only and do not necessarily reflect those of the European Union or the REA. Neither the European Union nor the REA can be held responsible for them.

## Author contributions
J.L. conceived of the idea and formulated the perspective together with G.F.V. J.L. conducted the literature review and wrote the first manuscript draft. G.F.V. and K.M.J. critically commented on the manuscript and added and/or revised content.

## Competing interests
The authors declare no competing interests.
