## [Transparent Peer Review file · Communications Earth & Environment]

Towards multitrophic soil biogeochemistry: how to conceptually include soil fauna in carbon cycling

Corresponding Author: Dr Justine Lejoly

This manuscript has been previously reviewed at another journal. This document only contains information relating to versions considered at Communications Earth & Environment.

Version 0:

Decision Letter:

Dear Dr Lejoly,

Your manuscript titled "Towards multitrophic soil biogeochemistry: how to conceptually include soil fauna in carbon cycling" has now been seen by 3 reviewers, whose comments are appended below. You will see that they find your work of some potential interest. However, they have raised quite substantial concerns that must be addressed. In light of these comments, we cannot accept the manuscript for publication, but would be interested in considering a revised version that fully addresses these serious concerns.

We hope you will find the reviewers' comments useful as you decide how to proceed. Additionally, the following editorial thresholds should be met in the revised manuscript:

* Provide novel and critical synthesis of knowledge towards the concept of adding soil fauna in carbon cycling. Also clearly explain how the present article is suggesting new concepts compared to Angst et al. (2024, Nature Communications) and Bonfanti et al. (2025, Functional Ecology).

* Enhance the paper's coherence and impact by either focusing on a comprehensive synthesis of existing knowledge and a proposal of a new conceptual framework.

If additional work allows you to either incorporate or refute these criticisms, we will be happy to look at a substantially revised manuscript. If you choose to take up this option, please either highlight all changes in the manuscript text file, or provide a list of the changes to the manuscript with your responses to the reviewers.

When resubmitting, please provide a point-by-point response to the reviewers' comments. Please submit your responses as a separate file, distinct from your cover letter where you can add responses to the Editors' comments that you do not want to be made available to the reviewers. Word files are preferred. We recommend that any figures, tables or graphs that are included in the response to reviewers are also included in the main article or Supplementary Information.

If the revision process takes significantly longer than three months, we will be happy to reconsider your paper at a later date, as long as nothing similar has been accepted for publication at Communications Earth & Environment or published elsewhere in the meantime.

Please use the following link to submit your revised manuscript, point-by-point response to the reviewers' comments with a list of your changes to the manuscript text (which should be in a separate document to any cover letter), a tracked-changes version of the manuscript (as a PDF file) and any completed checklist:

Link Redacted

Please do not hesitate to contact us if you have any questions or would like to discuss the required revisions further. Thank you for the opportunity to review your work.

Best regards,

Tiago Ferreira
Editorial Board Member
Communications Earth & Environment
orcid.org/0000-0002-4088-7457

EDITORIAL POLICIES AND FORMAT

If you decide to resubmit your paper, please ensure that your manuscript complies with our editorial policies and complete and upload the checklist below as a Related Manuscript file type with the revised article:

- Behavioural and social science
- Ecological, evolutionary & environmental sciences
- Life sciences

For your information, you can find some guidance regarding format requirements summarized on the following checklist: (<https://www.nature.com/documents/commsj-phys-style-formatting-checklist-article.pdf>) and formatting guide (<https://www.nature.com/documents/commsj-phys-style-formatting-guide-accept.pdf>).

REVIEWER COMMENTS:

Reviewer #1 (Remarks to the Author):

The review by Lejoly et al. emphasizes the neglected role of soil fauna in SOM dynamics and the formation of POM and MAOM. The topic is important and timely, and the manuscript generally written well. My strongest concern with the manuscript, however, is its novelty. Recent reviews, i.e., those by Angst et al. 2024 Nature Communications and Bonfanti et al. 2025 Functional Ecology, which the authors cite, comprehensively review the topic, and it seems that the current manuscript mixes both papers without coming up with substantially new points or concepts. Moreover, with respect to the relevance of soil fauna for SOM dynamics, the manuscript often remains less specific than the above papers.

Line comments:

L69-78 This passage could be more nuanced and/or informative. Why can SOM dynamics be better understood when separating soil into POM and MAOM?

What is the typical persistence of POM and MAOM in soil? Via which pathways do both fractions form? Moreover, microbial necromass is only persistent if part of MAOM. In coniferous forests, for example, the "plant" pool of persistent SOM is typically much larger than the persistent necromass pool. I recommend writing this passage in a more nuanced way.

L78/79 The authors mix terms here. What is persistent, what is stabilized? Some authors have recently advised against using the term "stabilization" with respect to SOM or POM and MAOM.

L79 ff. I wouldn't say that abiotic factors are the major control for MAOM formation; it's rather a combination of biotic and abiotic factors, i.e., MAOM in boreal forests is not low because temperatures are low per se, but because low temperatures hamper microbial activity.

L87 Accelerated decomposition rather reduces POM

L121/122 verb is missing

L126 ff. Is this a hypothesis by the authors? If so, this could become clearer.

L207/208 Sentence not entirely clear

L233 Where is C stored as POM, in humus or the mineral soil?

Box 1 So is the humus form dependent on soil fauna or does the humus form select for certain taxa? If the former, Box 1 does not entirely fit to the title of this section.

L245 What do they mineralize and why does this reduce SOM formation and stabilization?

L255 Are there reports on the negative effects of microbivores on SOM? Above, the authors also state that microbivores could increase microbial necromass production, which could potentially increase MAOM formation. Argumentation inconsistent.

L264-266 Why? How could plasticity or omnivory influence SOM compared to more specific feeding strategies?

L271 Subject-verb disagreement

Table 2: What is the difference between SOM stabilization and MAOM formation or C protection in aggregates?

Table 3: What is SOM formation? In general, the authors appear to use SOM formation and SOM stabilization interchangeably. What do they refer to with the two terms?

Caption of Fig. 1 lacks detail, e.g., but not limited to, what do the different arrows and colors indicate? What do the water drops indicate?

Fig. 2: What about abiotic constraints such as pH? Soil type in the figure only appears to refer to texture. Moreover, how does soil type constrain fauna? The figure indicates that soil type determines the amount of POM and MAOM only. The sandy soil rather appears "rocky" than sandy. Where do I see the humus forms in the figure? It is not clear to me what the bulk soil box indicates. Rhizosphere: root turnover can also be a "source" of POM. Where is POM in the detritusphere and consumption and excretion panels?

Reviewer #2 (Remarks to the Author):

This work presents a substantial synthesis of over 170 scientific articles of various types on a subject at the intersection of two disciplines: soil biology and biogeochemistry. It paves the way for the emergence of new research fields. The thematic focus of this work is therefore highly relevant, and its contribution in terms of knowledge synthesis and conceptual framework is highly valuable.

The text is globally well-written and very clear. Nevertheless, I was slightly confused about how to categorize this manuscript, as it oscillates between a complete synthesis and the proposal of a new conceptual framework. Indeed, this review is likely insufficiently documented to claim an exhaustive synthesis of current knowledge and the latest advancements in the field. Simultaneously, it does not present new scientific concepts but rather a re-evaluation of existing ones.

Given the immense diversity of soil organisms, it is sometimes challenging to identify major trends and move beyond simple experimental studies to understand the emergent properties of this complex system and its underlying processes. This work, however, has boldly attempted to do so. The role of microorganisms and the interactions between fauna and microorganisms are well-synthesized and have generally been the most documented, reflecting perhaps the research field that has progressed the most in recent years. Other aspects concerning macrofauna are interesting but sometimes insufficiently documented, which likely reflects both the available literature and the significant diversity of form and function of these organisms. Nevertheless, certain points are clearly insufficiently supported by existing literature; comments have been provided to easily improve these points.

In summary, I give a favorable opinion, subject to the realization of some minor corrections indicated in the comments below.

Best regards,

Reviewer #3 (Remarks to the Author):

This is nice review about soil fauna effect on carbon cycle. I like distinction on trophic and non-trophic partway, however while the traffic pathway is well elaborated, pathway will deserve more deeper elaboration, even using literature which was already mentioned.

As mentioned in 140, fauna activity, my altar CN ratio, of DOM and doing so they might affect activity of microbes causing positive or negative priming effect. This was also empirically tested by Frouz et al, 2020 Applied Soil Ecology 153:103585.

Bioturbation in particular, should receive much larger attention, here important is to mention

that bioturbation beside its immediate effect on microbial activity has long lasting legacy effect which may alternate functioning of food web and as was correctly mentioned fauna activity depends on environmental framework but in some degree fauna bioturbation alter environmental framework which may cause for example change in soil food web composition, these legacy effects should be at least briefly mentioned, see Frouz 2024 Soil Biology and Biochemistry 189, 109289 where more relevant sources can be found.

** Visit Nature Portfolio's author and referees' website at www.nature.com/authors for information about policies, services and author benefits**

Communications Earth & Environment is committed to improving transparency in authorship. As part of our efforts in this direction, we are now requesting that all authors identified as 'corresponding author' create and link their Open Researcher and Contributor Identifier (ORCID) with their account on the Manuscript Tracking System prior to acceptance. ORCID helps the scientific community achieve unambiguous attribution of all scholarly contributions. You can create and link your ORCID from the home page of the Manuscript Tracking System by clicking on 'Modify my Springer Nature account' and following the instructions in the link below. Please also inform all co-authors that they can add their ORCIDs to their accounts and that they must do so prior to acceptance.

Version 1:

Decision Letter:

Dear Dr Lejoly,

Your manuscript titled "Towards multitrophic soil biogeochemistry: how to conceptually include soil fauna in carbon cycling" has now been seen by our reviewers, whose comments appear below. In light of their advice we are delighted to say that we are happy, in principle, to publish a suitably revised version in Communications Earth & Environment. Please consider the following point in during revision.

* We noticed there are not many references from 2024-2025. Given that some of the ground has already been covered by the earlier two reviews [Bonfanti et al. (2025) and Angst et al. (2024)], please update the reference list and include more recently published papers in this area of research.

We therefore invite you to revise your paper one last time to address the remaining concerns of our reviewers. At the same time we ask that you edit your manuscript to comply with our format requirements and to maximise the accessibility and therefore the impact of your work.

EDITORIAL REQUESTS:

****Please take care to match our formatting and policy requirements. We will check revised manuscript and return manuscripts that do not comply. Such requests will lead to delays. ****

SUBMISSION INFORMATION:

OPEN ACCESS:

Communications Earth & Environment is a fully open access journal. Articles are made freely accessible on publication. For further information about article processing charges, open access funding, and advice and support from Nature Portfolio, please visit <https://www.nature.com/commsenv/open-access>

Link Redacted

**** This url links to your confidential home page and associated information about manuscripts you may have submitted or be**

reviewing for us. If you wish to forward this email to co-authors, please delete the link to your homepage first **

Best regards,

Somaparna Ghosh, PhD
Associate Editor,
Communications Earth & Environment
Consulting Editor,
Communications Sustainability

REVIEWERS' COMMENTS:

Reviewer #1 (Remarks to the Author):

While the authors have made efforts to address my concerns, I am still not convinced about the novelty of the work. For example, the authors claim that feces or carcasses have not been covered by Bonfanti et al. or Angst et al., which is wrong. The trophic pathways referred to in the manuscript have been covered by either Bonfanti et al. or Angst et al., including (microbial) decomposition pathways and processes beyond the formation of SOM. While I see some novel aspects in the manuscript, I do not believe that they warrant publication as a full-length review.

Reviewer #3 (Remarks to the Author):

I am fine with changes made

** Visit Nature Portfolio's author and referees' website at www.nature.com/authors for information about policies, services and author benefits**
